# Bias-Modulated High Photoelectric Response of Graphene-Nanocrystallite Embedded Carbon Film Coated on n-Silicon

**DOI:** 10.3390/nano9030327

**Published:** 2019-03-01

**Authors:** Xi Zhang, Zezhou Lin, Da Peng, Dongfeng Diao

**Affiliations:** Institute of Nanosurface Science and Engineering, Guangdong Provincial Key Laboratory of Micro/Nano Optomechatronics Engineering, Shenzhen University, Shenzhen 518060, China; zh0005xi@szu.edu.cn (X.Z.); zezhoulin@163.com (Z.L.); pandavmac@163.com (D.P.)

**Keywords:** graphene nanocrystallite, bias-modulated Fermi level, carbon film, photoelectric response, tunable spectral response

## Abstract

We propose that bias-modulated graphene-nanocrystallites (GNs) grown vertically can enhance the photoelectric property of carbon film coated on n-Si substrate. In this work, GN-embedded carbon (GNEC) films were deposited by the electron cyclotron resonance (ECR) sputtering technique. Under a reverse diode bias which lifts the Dirac point of GNs to a higher value, the GNEC film/n-Si device achieved a high photocurrent responsivity of 0.35 A/W. The bias-modulated position of the Dirac point resulted in a tunable ON/OFF ratio and a variable spectral response peak. Moreover, due to the standing structured GNs keeping the transport channels, a response time of 2.2 μs was achieved. This work sheds light on the bias-control wavelength-sensitive photodetector applications.

## 1. Introduction

Photoelectric sensors are widely used in optical communication, infrared ranging, environmental monitoring, military security inspection, biomedical imaging, and scientific research [1,2,3,4]. Graphene is a promising candidate for new-generation photodetectors due to its outstanding photoelectric properties [5,6]. The combination of graphene and silicon, with their known conductive behaviors, offers new unique properties and draws great research attention [7,8]. However, plane-graphene photodetectors present low photocurrent responsivity due to the lack of trapping centers in plane-graphene, which causes the rapid recombination of photoexcited carriers. Large-scale graphene with few defects is difficult to obtain using modern techniques such as chemical vapor deposition (CVD) [9,10,11,12,13,14], reduced graphene oxide (RGO) [15], and mechanical lift-off techniques [16]. The effective areas of the reported graphene/silicon photodetector are usually tens of μm^2^. Moreover, the transfer technique for the graphene/silicon heterojunction is complicated and limits the material’s industrial applications. Therefore, a method to mass-produce a photodiode with low cost and high responsivity is highly desired.

Amorphous carbon (a-C) thin films were synthesized from coal as a solid carbon source [17]. The a-C thin film has an optical transmittance of >96% in the spectral range from 350 nm to 900 nm, and it can be transferred to various substrates. In 1996, Yu prepared a carbonaceous thin-film/n-type silicon layer photovoltaic cell. It was confirmed that the carbonaceous film/n-type silicon junction was a heterojunction [18]. The a-C film is the earliest carbon-based material applied in heterojunctions for photoelectric devices with a responsivity of 0.18 A/W [19]. In 2012, Xue et al. tested the substrate resistivity [20], light-induced resistance effect [21], and photoconductivity [22] of Pd-doped amorphous carbon film/SiO_2_/Si heterojunctions. The device size of the a-C film photodetector can be as large as a commercial photodetector (~tens of mm^2^). However, the responsivity of a-C film is still rather low.

Recently, we proposed a method for the direct preparation of graphene nanocrystallite-embedded carbon (GNEC) film via a low energy electron irradiation technique [23]. Unlike a-C film, GNEC film contains many vertically grown graphene nanocrystallites (GNs). It exhibits a good magnetic and triboelectric performance [23,24] due to the large amount of graphene edges in GNEC film, which serve as trapping centers for electrons and net spin. Hence, it is expected that GNEC film can be applied in a photoelectric response area, since the edge electron trapping centers may assist with photocurrent generation in the photovoltaic process.

In this paper, we report the photo-electric behavior of GNEC films coated on n-Si substrates prepared by electron cyclotron resonance (ECR) plasma sputtering technology under bias voltage modulation. The external bias could modulate the Fermi level (*E*_F_) of GNs and help to form a p–n junction with n-Si. The photoelectric properties of the GNEC film/n-Si heterojunction were tested, and the mechanism of the nanostructure’s effect on the photoelectric properties of the GNEC coating was unveiled.

## 2. Experimental Methods

The GNEC films were coated on n-Si using an ECR plasma sputtering system. A detailed description of the sputtering system was reported in our previous works. A 500-W microwave was delivered into the vacuum chamber to generate the plasma at an argon pressure of 4 × 10^−2^ Pa. The mirror confinement magnetic field was applied in order to enhance the plasma density. A negatively biased glassy carbon target is known to attract Ar^+^ ions to sputter carbon species towards the substrate [25]. The substrate was a single-sided polished 4-inch N-type conductive silicon wafer (square resistance of ~10 Ω, 0.5 mm thickness). A deposition time of 30 min was adopted in this work to control the thickness to be around 70 nm. A positive deposition bias (*V*_dep_) was applied on the substrate to attract electrons [26]. With the assistance of the low-energy electrons, GNs grow perpendicularly to the substrate. We prepared films under *V*_dep_ = 20, 50, and 80 V. The nanostructures of the carbon films were analyzed using Raman spectroscopy (HORIBA, HR-Resolution; wavelength of 532 nm) and transmission electron microscopy (TEM, JEOL, JEM-3200FS).

A finger-type 50-nm-thick gold electrode was deposited on top of the GNEC film using UV lithography [27]. A 50-nm-thick flat gold electrode was deposited on the back of the silicon substrate. The Lakeshore TTPX cryogenic probe station covered by a built-up darkroom was used as the electrode contact. The optical setup was also placed in the built-up darkroom. Free-space lasers were used as the light source with a spot size of approximately 0.2 mm^2^. A Keithley 4200-scs semiconductor characterization analyzer was used to measure the I–V curves. To measure the frequency response, a diode laser (Coherent OBIS, λ = 785 nm) was modulated ON/OFF with a 50% duty cycle by a 100-kHz square waveform generated by a function generator (Keysight 33,600 A). The rise/fall time of the modulated diode laser was 10 ns. The spectral response was measured by a white light source, a monochromator, a chopper, and a lock-in amplifier. A xenon light in the visible to near-infrared (300 nm to 1100 nm) range was used as the white light source. The xenon lamp light was dispersed by a monochromator (precision of 0.5 nm) and the average power was in the order of 10 μW/nm.

## 3. Results and Discussion

### 3.1. Nanostructure Characterization of GNEC Films

The carbon films were coated on the n-Si substrate using an ECR sputtering system, as shown in Figure 1. The thickness of the GNEC film on n-Si was about 70 nm, as measured by a profilometer (Bruker, Dektak-XT, AZ, USA). The coated wafer was cut into 5 × 5 mm^2^ square pieces. A finger-shaped electrode was deposited on the surface of the GNEC film by UV lithography, and a 50-nm flat gold film was deposited on the back of the Si substrate. The finger-type mask was designed with a central spine (~300 μm width) and two sides of branches (~150 μm width). The spacing between the two branches was ~500 μm. The spine and branches were all designed in the shape of wedges to collect the electrons efficiently. High-resolution transmission electron microscopy (TEM) was used to explore the cross-section sample of the 50-V film, as shown in Figure 1c. A cross-section sample with a lateral thickness of 100 nm was prepared by focused ion beam (FIB) after a protection layer was deposited on top (see Appendix A). The minimal thickness was more conducive to our observation of the vertical growth structure of GNs under TEM. The etching energy was high during FIB deposition, so it was necessary to deposit a 2 μm protective layer (platinum material) on the surface of the sample to prevent damage to the surface morphology during the FIB etching process and TEM observation. Although an overlay image of a 100-nm lateral thickness cross-section was revealed during the TEM observation, as shown in the inset in Figure 1a, the tendency of GNs to grow perpendicular to the substrate was conspicuous.

Transmission electron microscopy (TEM) and Raman spectra of carbon films coated under ECR low-energy electron irradiation at *V*_dep_ = 20 V, 50 V, and 80 V are shown in Figure 2a–c. The red and green boxes in Figure 2a–c mark the areas with and without GN, respectively, and the insets are their fast Fourier transform (FFT) images [28,29]. The FFT images demonstrate that inside the GN two Laue spots appear, corresponding to the (0001)* facet of the multilayer graphene [30]. The reciprocal lattice distance is measured as d*_(0001)_ = 5.98/2 nm^−1^ = 2.99 nm^−1^. Thus, the interplanar distance can be calculated as d = 1/2.99 nm = 0.334 nm, in accordance with the (0001) facet distance of multilayer graphene. In the green square area, the FFT image does not show a pattern, indicating no GN formation. In the TEM image, we used white circles to indicate the boundaries of the GN on the GNEC film. The average size of the white circles indicates the width of the GN boundary. At *V*_dep_ = 20 V, the carbon film has an a-C structure. At *V*_dep_ = 50 V, the average size of the GN boundary is ~4 nm. At *V*_dep_ = 80 V, the average size of the GN boundary is increased to ~15 nm.

Raman spectra of films at *V*_dep_ = 20 V, 50 V, and 80 V are shown in Figure 2d–f. D, G, and 2D peaks were decomposed by Lorentz function fitting [31,32,33,34]. At *V*_dep_ = 20 V, the D, G, and 2D profiles are vague and wide, and the intensity ratio between D and G peaks (I_D_/I_G_) is almost 1, indicating the formation of a-C structures. At *V*_dep_ = 50 V, a stronger 2D peak around 2700 cm^−1^ appears and the I_D_/I_G_ increases to 1.34, indicating the formation of small-size GNs in the carbon film. At *V*_dep_ = 80 V, a sharp 2D peak can be observed and I_D_/I_G_ increases to 2.25. This indicates that, with the increase of low-energy electron radiation energy, the structure of GN on GNEC film tends to be ordered and the width of the GN boundary tends to increase.

Figure 2d–f also show the red shift of the 2D peak position versus *V_dep_*. *ω* decreases from 2693 to 2688 cm^−1^ and then to 2683 cm^−1^ as *V_dep_* increases from 20 to 50 V and then to 80 V. The shift of the 2D band can be explained by the edge quantum effect theory [35,36]. This theory clarifies the correlation, Δω∞(*E_K_/μ*)^1/2^/(*d_k_K*), among the shift of the 2D peak (Δω) from the ω(Bulk) of large graphene bulk to the nanocrystallite ω(*K*), the size *K*, the average bond length (*d_K_*), the average bond energy (*E_K_*), and the reduced mass of the dimer atoms (*μ*). Thus, as the crystallite size increases, the average bond length expands and the average bond energy drops, leading to the decrease of ω towards the reference point of large graphene. Thus, the trend of the red shift of ω(*K*) as *V_dep_* increases agrees well with the theoretical prediction, verifying the increase of the nanocrystallite size of GNEC film.

### 3.2. Device Characterization and Nanostructure-Dependent Photocurrent

V curves in darkness and at a power (P) of 10 or 40 mW at a wavelength of 785 nm laser are shown in Figure 3a–c. The dark currents showed rectification characteristics. Since the film with *V*_dep_ = 20 V had an amorphous structure, a low photocurrent was generated under reverse bias. At a diode bias of −5 V, the GNEC film at *V*_dep_ = 50 V produced a larger photocurrent of 9 × 10^−3^ A. As the GN size increased, the GNEC film at *V*_dep_ = 80 V produced the highest photocurrent of 1.44 × 10^−2^ A at reverse *V*_diode_ = −5 V. Increasing the irradiation voltage (*V*_dep_) enhanced the crystallization of the GNEC film, leading to a larger photocurrent under reverse bias. Photocurrents are extremely low at zero bias and increase drastically with reverse bias. Graphene/n-Si [37] also exhibited a drastically photocurrent depression at *V*_diode_ = 0 V due to the Dirac cone of graphene. This behavior at zero bias is largely different from that of the GNEC/p-Si device [38]. The GNEC film/p-Si showed a considerably large photocurrent at *V*_diode_ = 0 V due to the natural p–n junction formation between the electron-trapped GNs and p-Si. In contrast, in GNEC/n-Si, the reverse bias could depress the GNs’ Fermi level to open up available energy states for holes to inject. Under illumination, the flux of photocurrent was limited by the density of states of graphene nanocrystallites near the Fermi level. Thus, the minimum photocurrent points in Figure 3 roughly indicate the location of the minimum density of states points. It decreased from −0.12 to −0.17 V and further to −0.2 V as the V_dep_ increased from 20 to 50 V and finally to 80 V (see Appendix A). This is because the more crystallized 80 V film formed a stronger p–n junction with n-Si and the relative position of E_F_ had a deeper location. A stronger photocurrent under illumination was generated under reverse bias. Photocurrents almost reached saturation at *V*_diode_ = −2 ~ −3 V. As the illumination power decreased from 40 to 10 mW the saturation voltage decreased, since weak light can be easily used by the device while reverse bias cannot make out a further product. The diode bias-induced tunability of the relative positions of the Fermi levels resulted in a tunable ON/OFF ratio. The ON/OFF ratio of the 80 V film device varied from 3.2 at −0.2 V to 21.4 at −2 V. The application of a reverse *V*_diode_ raised the Fermi level (E_f_) of GN, inducing the p-type behavior of GN and forming a larger junction with n-Si.

### 3.3. Nanostructure-Dependent Spectral Responsivity

As shown in Figure 4, the spectral responsivity curves of the 50 and 80 V GNEC film devices at different values of V_diode_ were obtained. The GNEC/n-Si showed a broadband photo-detection ability and the peak position was located at the near-infrared region. The 80 V GNEC film showed a higher responsivity of 0.35 A/W compared to that of the 50 V GNEC film (0.27 A/W) under a 900 nm laser, due to the increasing crystallization of GNs in the 80 V film. Since the illumination power of the white light source was rather weak (~10 μW/nm), the R difference between −1 and −2 V was not as large as that in Figure 3. This is because the saturation voltage is quite low under weak light. As the bias decreased to 0 V, the peak responsivities decreased to low values (2.42 mA/W for the 50 V film and 0.12 A/W for the 80 V film). It can also be seen that the peak position blue-shifted with reverse bias, increasing from 0 to −2 V (from 1035 to 900 nm of the 80 V film). This is because the applied reverse bias can raise the Dirac point of the GN to a higher value, and increase the potential barrier of the p–n junction interface. Thus, the electron excitation requires a larger photon energy, leading to the blue-shift phenomenon. The bias-dependent response peak not only confirms the bias-modulation to the Dirac point of the GNs, but also sheds light on the tunable spectral sensitivity applications of GNEC/n-Si devices.

### 3.4. Response Time

Generally, the increase in light responsivity is accompanied by an increase in the light response time [39]. Unlike other materials, a large number of GNs on GNEC membranes play a crucial role in charge transport. GN grows vertically on a substrate, increasing the charge transport channels, thereby increasing the light response speed. As shown in Figure 5a,b, the optical response speed of the GNEC film device was tested under square wave frequencies of 1 and 100 kHz. The response time of the GNEC film/n-Si device was 2.2 μs, enabling commercial fast-response applications such as high-speed imaging, barcode scanners, and photoelectric encoders. In addition, the GNEC film/n-Si device exhibited a recovery time of 2.7 μs, owing to the fast carrier diffusion time induced by the electron doping during GNEC film deposition.

In contrast to plane-graphene, GNEC film is a directly deposited carbon film which contains a large amount of standing structured GNs. Although plane-graphene has massless Dirac Fermions and presents an ultrafast response [40], the device area is usually limited to tens of μm^2^ and a complicated transfer technique is needed. In contrast, the a-C film device area is usually tens of mm^2^ and no transfer is needed for industrial applications. We aim to enhance the photoresponsivity of such devices by introducing standing structured GNs into the film. Despite its large area of 5 × 5 mm^2^, the performance of our device was outstanding compared with some other small-area plane-graphene devices. In comparison with the previously reported graphene photodetectors, we summarize some features of the previously reported photodetectors in Table 1. The device developed here showed a high responsivity (0.35 A/W) compared with a traditional a-C film device (0.18 A/W). In addition, the response time was reduced to 2.2 μs. The edge of the nanosheet acted as electron capture centers. Under reverse bias, the device exhibited bias modulation characteristics. The GNEC film prepared by ECR technology thus has the characteristics of low cost, easy preparation, and large-area production.

### 3.5. Discussion of Principle

Under the irradiation of low-energy electrons, a large amount of standing structured GNs are vertically grown to provide a high density of edges for the GNEC film. Figure 6a illustrates the energy band diagram, showing the Fermi levels of GN (*E*_f_ (GN)) and n-Si (*E*_f_ (Si)) at thermal equilibrium in darkness. The Dirac cone shape on the left is an illustration of the band of GNs. In the film, it may not necessarily be that shape. The Dirac point represents the minimum density of states point. The edge states of GNs tend to capture electrons and raise the *E*_f_ (GN) [45]. The two curves on the right represent the top of the valence band (*E*_V_) and the bottom of the conduction band (*E*_C_) of n-Si. The *E*_f_ (Si) of n-Si locates near the valence band (*E*_V_) and bends at the interface with a potential barrier *q*V_d_ due to the formation of a p–n junction. As shown in Figure 6b, when reverse bias voltage is applied, quasi Fermi levels—separately for holes of n-Si (*E*_f_ ‘(Si)) and electrons of p-GN (*E*_f_ ‘(GN))—are formed in the non-equilibrium situation. An applied reverse bias raises the Dirac point of GNs, increasing the chemical potential disparity between GNs and n-Si and leading to the formation of junctions [46]. The potential barrier was increased by *q*V_r_, preventing the dark current. The applied reverse bias can raise the Dirac point of GNs to a higher value, opening up a large number of accessible states for the holes to inject into and allowing a complete collection of the injected holes. Under illumination, photo-excited holes tend to flow into the accessible GN states. Hence, the electron-hole recombination rate is drastically reduced by GN trapping centers, leading to an outstanding responsivity.

There is huge potential for GNEC film to further enhance its responsivity, for example, by increasing the density of GN edges, preventing dark current leakage by introducing an interface insulator [3], nitrogen-doping, or other dopants. Besides, increasing the substrate beyond 80 V would increase the crystallization of the GNs and bring the film towards a graphite-like structure. Technically, excessive substrate bias may cause anomalies in the deposition equipment. Physically, increasing the deposition bias would probably enhance the performance of the device, but there should be a critical point somewhere larger than 80 V. At that critical point, the graphite-like film would lack trapping centers and the photoelectric performance would decrease.

## 4. Conclusions

GNs vertically grown in a GNEC film can prominently improve the photoelectric performance of the material by improving its ability to collect photogenerated electrons and prolonging the lifetime of photoexcited carriers. The GNEC film/n-Si heterojunction exhibited a high photocurrent responsivity of 0.35 A/W under reverse bias, compared with the responsivity of 0.18 A/W of a-C film/n-Si. In the GNEC film preparation process, we can control the crystallinity by adjusting the energy of low-energy electron irradiation, so that the photoelectric performance of the GNEC film/n-Si photodetector can be improved.

The GNEC film coated on n-Si exhibited excellent photocurrent responsivity (0.35 A/W) and a highly sensitive response time (τ_rise_ = 2.2 μs) under reverse bias. The excellent photoelectric performance exhibited under reverse bias was attributed to the GN crystallization and the increased Dirac cone. The bias-modulated position of the Dirac point resulted in a tunable ON/OFF ratio and a variable spectral response peak (from 900 to 1035 nm), enabling bias-control wavelength-sensitive photodetector applications. The GNEC film/n-Si heterojunction photodetector allows low cost, large area, and efficient production. It can be used for a variety of tunable photoelectric devices with high responsivities over broadband, having a broad application potential in industrial production.

## Figures and Tables

**Figure 1 nanomaterials-09-00327-f001:**
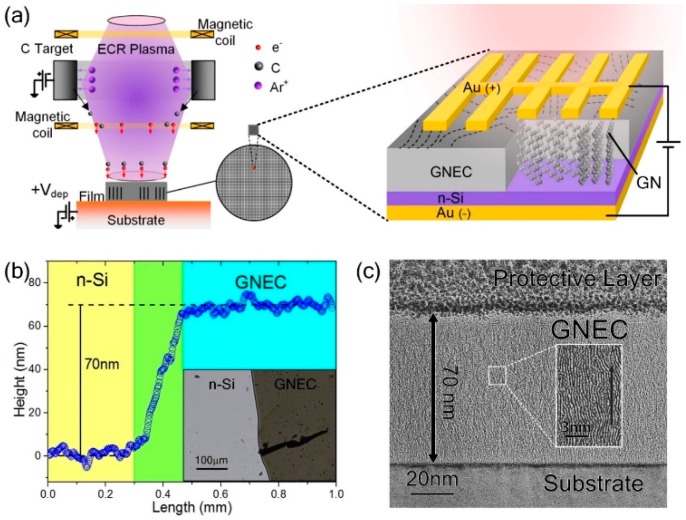
(**a**) A schematic setup of the electron cyclotron resonance (ECR) plasma low-energy electron deposition. The inset shows standing structured graphene nanocrystallite (GN) induced by low-energy electrons irradiation (modulated by *V*_dep_) in graphene nanocrystallite-embedded carbon (GNEC) film. Diode bias (*V*_diode_) is applied on the GNEC film against Si. (**b**) The stepped profile of GNEC film coated on n-Si. (**c**) High-resolution TEM image of the cross-section sample of the 50-V film prepared by focused ion beam (FIB) deposition.

**Figure 2 nanomaterials-09-00327-f002:**
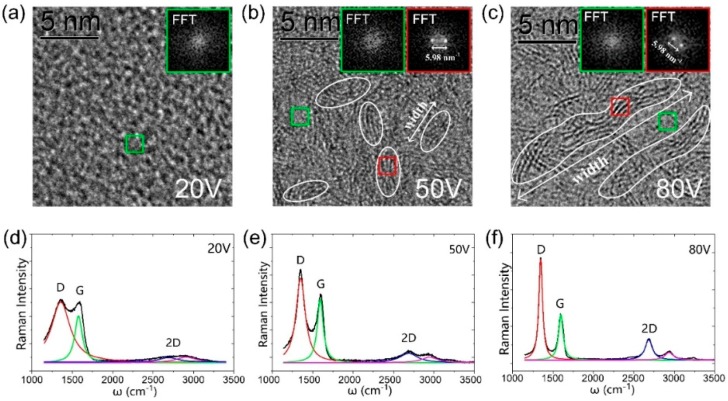
High-resolution TEM plan-view images of GNEC films deposited at a *V*_dep_ of (**a**) 20 V, (**b**) 50 V, and (**c**) 80 V. Insets are the fast Fourier transform (FFT) images of the selected region (green and red squares). As the *V*_dep_ increases, the GN boundary width increases. Raman spectra of GNEC films with a *V*_dep_ of (**d**) 20 V, (**e**) 50 V, and (**f**) 80 V. As the *V*_dep_ increases, D, G, and 2D peaks are all sharpened.

**Figure 3 nanomaterials-09-00327-f003:**
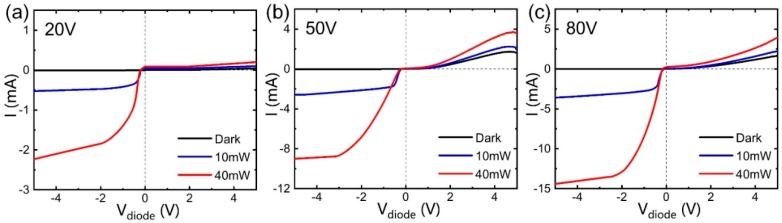
Current–voltage (I–*V*_diode_) curves under darkness and illumination (P = 10 or 40 mW) of 785 nm laser of carbon films/n-Si with a *V*_dep_ at (**a**) 20 V, (**b**) 50 V, and (**c**) 80 V. The GNEC film/n-Si exhibits a large photocurrent under reverse *V*_diode_. As the reverse bias voltage increases from 0 to −2 ~ −3 V, the photocurrent tends to saturate.

**Figure 4 nanomaterials-09-00327-f004:**
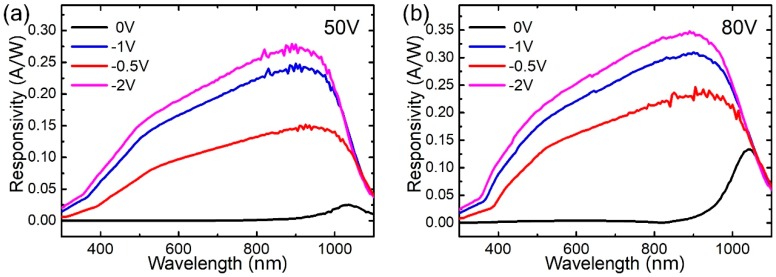
(**a**) The variation of the photocurrent responsivity (*V*_dep_ = 50 V) in the wavelength range from 300 to 1100 nm under different bias voltages (*V*_diode_ = 0 V, −0.5 V, −1 V, −2 V). (**b**) The variation of the photocurrent responsivity (*V*_dep_ = 80 V) in the wavelength range from 300 to 1100 nm under different bias voltages (*V*_diode_ = 0 V, −0.5 V, −1 V, −2 V). Applying a reverse bias voltage will cause a blue shift in the response peak.

**Figure 5 nanomaterials-09-00327-f005:**
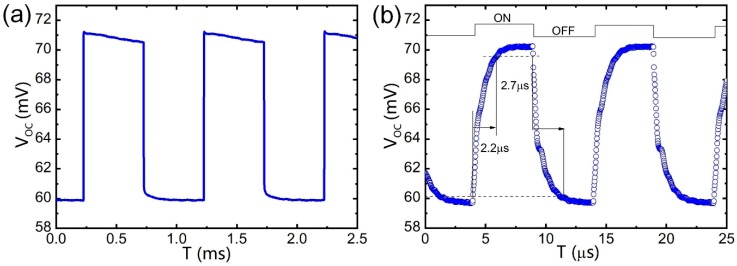
(**a**) Response time of the GNEC film/n-Si photodetector under 1 kHz square wave optical signal. (**b**) Response time of the GNEC film/n-Si photodetector under 100 kHz square wave optical signal. High sensitivity is demonstrated, the ON/OFF response rise time is 2.2 μs, and the recovery time is 2.7 μs.

**Figure 6 nanomaterials-09-00327-f006:**
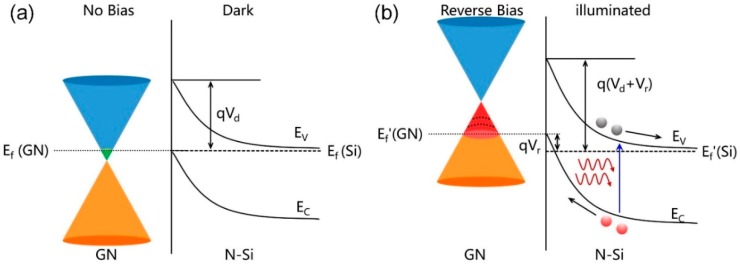
(**a**) Energy band structure diagram of the GNEC film/n-Si heterojunction at zero bias in the dark state. (**b**) Energy band structure diagram of the GNEC film/n-Si heterojunction under reverse bias in illumination.

**Table 1 nanomaterials-09-00327-t001:** Summary of device parameters of several typical 2D material photodetectors previously reported and our own device.

Type of Devices	Response Time	Responsivity	Wavelength
a-C film/Si [19,24]	-	180 mA/W	850 nm
MoS_2_-WS_2_ [41]	4 μs	4.36 mA/W	532 nm
Plasmon resonance Au nanoparticle–Gr [42]	50 μs	6 mA/W	532 nm
Gr–MoS_2_–Gr [43]	50 μs	220 mA/W	488 nm
Combined Monolayer Gr (MLG)/Si [44]	100 μs	29 mA/W	850 nm
GNEC/n-Si (this work)	2.2 μs	350 mA/W	850 nm

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
