# Peer review of "Bias-Modulated High Photoelectric Response of Graphene-Nanocrystallite Embedded Carbon Film Coated on n-Silicon"

_nanomaterials, 2019, doi:10.3390/nano9030327_

Reviewer 1 Report

The reviewer recommends this very good paper for publication in Nanomaterials with minor revisions.

- Improve English especially in the Introduction.

- Did you also measure the thickness with some methods, or did you infer it only from the deposition parameters of your system? (see line 64)

- Paragraph 3.5 - Discussion of principle - would be better presented with more details and comments.

Author Response

Thanks to the reviewer, please refer to the uploaded word for my response.

Reviewer 2 Report

The manuscript presents the authors' work on the fabrication and characterization of a graphene-nanocrystal-embedded carbon film/n-Si heterostructured device. The manuscript is well organized and their methods and analyses are clearly presented. The novelty of the device and the superiority of the device performance should be examined for the publication decision.

Clearly describe the novelty of the device in detail, with reviewing relevant past works.

I guess that the photocurrent responsivity and response time the authors obtained for their device may not be state-of-the-art values. For example see:

Y. Yao et al, "Wide wavelength tuning of optical antennas on graphene with nanosecond response time", Nano Lett. 14, 214, 2014.

Reviewing the past reports on similar structured devices and also for all the category of graphene-based devices, clearly discuss how your device performance is positioned good or not.

For the title, "Silicon" should not be capitalized for the consistency with other words.

Same thing for "Electron" in the abstract section.

Also, for some parts in the manuscript, I am not quite sure if the word choice and grammar are all right. (e.g., "nanocrystallines" -> nanocrystals, "GNs embedded carbon films" -> GN-embedded carbon films) I would recommend the authors to let the manuscript go through a professional native-English editing service.

Author Response

(The authors gave the same response as above.)

Reviewer 3 Report

The authors report about the processing of directed graphene-nanocrystallines embedded in a carbon film on n-type Si substrate. They investigate the influence of low energy electrons at different voltage drops during the sputter process to the microstructure of the processed layer. The layer formation has been characterized by TEM imaging. The microstructural findings have been verified by Raman spectroscopy, which verifies that the structures shown in TEM images are indeed distributed over larger regions of the layer.

Moreover the interaction of these layers with light have been studied. This was done by photocurrent measurements with different applied bias voltages to the formed heterojunctions. Furthermore, the light response of the structures and their response time was analyzed. In the discussion part the authors explain the experimental results with the common energy band models for the constituents of the heterojunction.

The manuscript is clearly written and follows a straight-forward logical structure. The results are well presented and the conclusion drawn from them clearly derived. Therefore, I recommend this manuscript for publishing.

However, I have some minor open questions and comments to the authors as listed below:

(1)   Was the natural thin oxide layer of the used silicon wafer removed before sputtering or is it possible to have an additional thin barrier of SiO2 in the system?

(2)   What are the advantages of the presented structure compared to conventional, e.g., Si-based, photodetector (materials)? Is there potential in the new material system to perform better than Si, e.g. higher responsivity as ~0.6 A/W or lower cost?

(3)   Page 1, Line 25/26: Could you rephrase the sentence starting with “The combination…” the meaning is not clear. I suppose it should be “the two materials with their known semiconductor properties offer new unique properties if both are combined” – am I right?

(4)   Especially if looking at Fig. 3, the question arises if the trend goes on if V_dep is further increased. Is there any technical limitation or a physical consideration to limit V_dep to a maximum of 80 V?

(5)   Regarding the temporal response test: Maybe the term “high speed” or “fast response time” (in the abstract) is a bit misleading, since, e.g. graphene/silicon detectors reach response times of a couple of 10 ns and Si-based detectors do as well.

(6)   Regarding Fig. 5 and the temporal response test: Have the authors checked the slew rate of the rectangular pulse to be much faster than the measured values. Especially if the setup connected to the pulse generator is not completely impedance match etc. even a very high slew rate produced by the generator can decreased when it arrives at the sample. In the latter case the true response time could be covered by a systematical error.

(7)   Grammar/Spelling: Page 1, Line 12: small case: “electron cyclone resonance”;

Page 2 ff.: please check for a space between number and unit;

Page 4; Line 166: “in accordance”

Page 5, Line 161: space between “Figure” and “3”

Page 6, capture of Fig. 4: the -0.5 V in “V_diode = “ is missing

Author Response

(The authors gave the same response as above.)

Reviewer 4 Report

The present manuscript is an interesting study on the photoelectrical response of graphene nanocrystals. However, some issues should be addressed before considering it for publication:

-In the introduction section (line 41), the authors talk about the "huge potential" of a-C for photoelectric devices. There are still many issues to be solved and improvements to be achieved to be competitive with other technologies... I would relax the claim.
-There is an excess of self-citation [24-30]. The authors should just select one of those references.
SEM images of the intermediate fabrication steps may be interesting to the readers.
-Did you observe any Raman shift? Can you comment on that?
-The authors based the explanation of the photocurrent modulation in the position of the Dirac point. Could you show I-V characteristics defining actually the Dirac point? Is there really a Dirac point in this polycrystalline material o rather a minimum conductivity point? Are there more than one of those points?
A table comparing responsivity and response time with other experimental and commercial approaches should be included.
Finally, I recommend thorough professional revision of the text. There are too many grammatical errors.

Author Response

Thanks to the reviewer, please refer to the uploaded word for my response.

Round  2

Reviewer 2 Report

Point 4: Also, for some parts in the manuscript, I am not quite sure if the word choice and grammar are all right. (e.g., "nanocrystallines" -> nanocrystals, "GNs embedded carbon films" -> GN-embedded carbon films) I would recommend the authors to let the manuscript go through a professional native-English editing service.

Response 4:

- These words have already been mentioned in other articles [8-11]. We carefully revise the manuscript to avoid grammatical errors.

Those articles are all you own. Crystalline is an adjective. As I Googled, nobody uses the word "graphene nanocrystallines", rather than yourselves.

I recommended the authors to let the manuscript go through a professional native-English editing service, and your answer "We carefully revise the manuscript to avoid grammatical errors." means a rejection. As other reviewers also pointed out, the manuscript contains a number of grammatical errors and careless typos. Do go to a native editing service.

Author Response

Thank you for your review. We have changed "nanocrystallines" to "nanocrystallites" and "GNs embedded carbon films" to "GN-embedded carbon films".
In addition, we conducted a native editing service. English edited manuscript and certification can be viewed in "Supporting information for reviewer".

Reviewer 4 Report

No further comments

Author Response

Thank you for your review. We conducted a native editing service. English edited manuscript and certification can be viewed in "Supporting information for reviewer".